# miRNA-378 Is Downregulated by XBP1 and Inhibits Growth and Migration of Luminal Breast Cancer Cells

**DOI:** 10.3390/ijms25010186

**Published:** 2023-12-22

**Authors:** Vahid Arabkari, David Barua, Muhammad Mosaraf Hossain, Mark Webber, Terry Smith, Ananya Gupta, Sanjeev Gupta

**Affiliations:** 1Discipline of Pathology, Cancer Progression and Treatment Research Group, Lambe Institute for Translational Research, School of Medicine, University of Galway, H91TK33 Galway, Ireland; v.arabkari@gmail.com (V.A.); davidcu09@gmail.com (D.B.); mosarafacme@gmail.com (M.M.H.); mark.webber@universityofgalway.ie (M.W.); 2Krefting Research Centre, Department of Internal Medicine and Clinical Nutrition, Institute of Medicine, University of Gothenburg, 40530 Gothenburg, Sweden; 3Department of Biochemistry and Molecular Biology, University of Chittagong, Chittagong 4331, Bangladesh; 4Molecular Diagnostic Research Group, College of Science, University of Galway, H91TK33 Galway, Ireland; terry.smith@universityofgalway.ie; 5Discipline of Physiology, School of Medicine, University of Galway, H91TK33 Galway, Ireland; ananya.gupta@universityofgalway.ie

**Keywords:** endoplasmic reticulum stress, endocrine resistance, XBP1, miR-378, breast cancer, unfolded protein response

## Abstract

X-box binding protein 1 (XBP1) is a transcription factor that plays a crucial role in the unfolded protein response (UPR), a cellular stress response pathway involved in maintaining protein homeostasis in the endoplasmic reticulum (EnR). While the role of XBP1 in UPR is well-characterised, emerging evidence suggests its involvement in endocrine resistance in breast cancer. The transcriptional activity of spliced XBP1 (XBP1s) is a major component of its biological effects, but the targets of XBP1s in estrogen receptor (ER)-positive breast cancer are not well understood. Here, we show that the expression of miR-378 and PPARGC1B (host gene of miR-378) is downregulated during UPR. Using chemical and genetic methods, we show that XBP1s is necessary and sufficient for the downregulation of miR-378 and PPARGC1B. Our results show that overexpression of miR-378 significantly suppressed cell growth, colony formation, and migration of ER-positive breast cancer cells. Further, we found that expression of miR-378 sensitised the cells to UPR-induced cell death and anti-estrogens. The expression of miR-378 and PPARGC1B was downregulated in breast cancer, and higher expression of miR-378 is associated with better outcomes in ER-positive breast cancer. We found that miR-378 upregulates the expression of several genes that regulate type I interferon signalling. Analysis of separate cohorts of breast cancer patients showed that a gene signature derived from miR-378 upregulated genes showed a strong association with improved overall and recurrence-free survival in breast cancer. Our results suggest a growth-suppressive role for miR-378 in ER-positive breast cancer where downregulation of miR-378 by XBP1 contributes to endocrine resistance in ER-positive breast cancer.

## 1. Introduction

Stressful conditions, such as hypoxia, acidosis, oxygen, and nutrient deprivation in the tumour microenvironment, impair the folding capacity of the endoplasmic reticulum (EnR) and trigger an evolutionary conserved pathway known as the unfolded protein response (UPR). Activation of UPR initiates the activation of three molecular sensors: activated transcription factor 6 (ATF6), PKR-like ER kinase (PERK), and Inositol-requiring enzyme 1 (IRE1) located in the endoplasmic reticulum membrane [1,2,3,4]. IRE1 is the most conserved signalling branch of the UPR and harbours kinase and endoribonuclease (RNase) activities within its cytosolic domain [5]. Upon activation, IRE1 forms higher oligomeric complexes, which leads to trans-autophosphorylation of its kinase domain, which in turn engages its RNase activity [2]. Interestingly, the functional outputs of active IRE1 are tailored to the stimulus via its RNase activity. This occurs through the combination of unconventional splicing of XBP1 mRNA and regulated IRE1-dependent decay of mRNA (RIDD) [6]. While splicing of XBP1 mRNA is a cytoprotective response to UPR, RIDD has revealed many context-dependent outcomes [6]. Spliced XBP1 (XBP1s) encodes a transcription factor and is a key mediator of the UPR. XBP1s can elicit a wide range of qualitatively and quantitatively distinct outputs that are determined by the physiological context [7,8,9].

Signals that increase protein production and secretion can activate the UPR in the absence of detectable EnR stress [10,11]. Indeed, estradiol (E2)-stimulation induces significant protein synthesis in breast cancer cells and elicits activation of the UPR, which is required for E2-mediated cell proliferation [12]. HyT-induced destabilisation of EnR-localized HaloTag (ERHT) protein robustly activates transient and non-apoptotic UPR, accompanied by gene expression signature reflecting estrogen signalling [13]. Inhibition of the IRE1-XBP1 axis abrogated the activation of the estrogen pathway by HyT-induced destabilisation of ERHT [13]. The expression of XBP1 is upregulated upon E2-stimulation, which, in turn, potentiates estrogen receptor (ER)-dependent transcriptional activity [14,15]. Ectopic expression of spliced XBP1 can bring about estrogen-independent growth and provide resistance to endocrine therapy [16]. Knockdown of XBP1 reduces the growth of cancer cells under a hypoxic environment and diminishes their ability to form tumours in NOD SCID mice [17]. Further, increased expression of spliced XBP1 is associated with poor outcomes of ER-positive breast cancer [18]. Targeting XBP1 expression by STF-083010 (small molecule inhibitor of IRE1) restores endocrine sensitivity to tamoxifen-resistant MCF7 cells in a xenograft model [19].

The XBP1s contributes to the development of endocrine-resistant breast cancer by upregulating the expression of BCL2, p65/RelA, NCOA3, RRM2, CDC6, and TOP2A [16,20,21,22,23,24]. It is likely that other yet unidentified XBP1-regulated genes contribute to ligand-independent growth and endocrine-resistant disease in ER-positive breast cancer. The miR-378a is located in the first intron of the peroxisome proliferator-activated receptor γ coactivator-1β (PPARGC1B) gene. Several other miRNAs with sequences similar to miR-378a but different genomic localisations have been discovered and named hsa-miR-378-b, c, d1, d2, e, f, g, h, i, j. In humans, miR-378a (hereafter referred to as miR-378) is the most expressed of all the miR-378 sequences. The processing of a hairpin RNA precursor present in the first intron of the PPARGC1B gene generates mature miR-378. The miR-378 plays an important role in many types of cancer [25]. Despite the wealth of knowledge about the function of miR-378 in human cancers, not much is known about the mechanisms regulating its expression. The expression of miR-378 and its host gene, PPARGC1B, is downregulated in tumour samples from breast cancer patients. We hypothesise that induction of UPR in the tumour microenvironment may play a role in reduced expression of miR-378 in breast cancer.

Here, we report that XBP1s downregulate the expression of miR-378 and PARGC1B (host gene of miR-378) during UPR. We observed an inverse relationship between the mRNA levels of XBP1 and PPARGC1B in the TCGA breast cancer dataset. Higher expression of miR-378 was associated with better overall survival (OS) in ER-positive breast cancer. The ectopic expression of miR-378 significantly suppressed cell proliferation, colony formation and migration of ER-positive human breast cancer cells. Further, miR-378 expression sensitised MCF7 cells to UPR-induced cell death and anti-estrogens. Analysis of separate cohorts of breast cancer patients showed that a gene signature derived from miR-378 upregulated genes was associated with improved overall and recurrence-free survival.

## 2. Results

### 2.1. Expression and Prognostic Value of the miR-378 in Subtypes of Breast Cancer

The miR-378a is located in the first intron of the PPARGC1B gene. We used TNMplot to determine the expression of PPARGC1B and miR-378 in normal and tumour tissue samples of the breast. We found a reduced expression of PPARGC1B and miR-378 in tumour samples as compared to tumour adjacent normal tissue (Figure 1A,B). Next, we determined the expression of PPARGC1B in publicly available datasets using Oncomine. We found that upon separating the tumours by stage (Bittner dataset GSE2109), the expression of PPARGC1B showed a significant downregulation with increasing tumour stage (Figure 1C). Further, the expression of miR-378 was downregulated in a panel of breast cancer cells as compared to non-transformed MCF10A cells (Figure 1D). Next, we determined the association between miR-378 expression and outcome in subtypes of breast cancer. We observed that increased expression of miR-378 was associated with improved overall survival (OS) in ER-positive breast cancer (Figure 2A), whereas the expression of miR-378 was associated with poor outcomes in triple-negative breast cancer (Figure 2B). The association of miR-378 expression with outcome in the HER2-enriched subtype of breast cancer was not statistically significant (Figure 2C).

Further, we found that increased miR-378 expression was associated with better OS in patients with metastatic (lymph node-positive) ER-positive breast cancer (Figure 2D).

### 2.2. miR-378 Is Downregulated during Conditions of UPR

The miR-378 is downregulated in tamoxifen-resistant as well as long-term estrogen-deprived MCF7 cells [26]. Knockdown of miR-378 provides resistance to tamoxifen-mediated suppression of cell proliferation [26]. To understand the upstream regulators of miR-378 expression, we analysed the gene expression profile (GSE63252: Total RNA was extracted from HCT116 cells treated with BFA or TM for 24 h, and microarray analysis was carried out using Affymetrix HG-U133_Plus-2 arrays) and found a significant downregulation of PPARGC1B during the conditions of EnR stress. Since most intronic miRNAs are co-transcribed along with their host gene, we reasoned that the expression of miR-378 might be repressed during UPR. The activation kinetics of three UPR sensors in response to stress induced by different pharmacological agents have shown fundamental differences in their recognition of different forms of EnR stress. Since microarray analysis was performed in colorectal cells, we used cell lines from colorectal (HCT116 and RKO) and breast cancer (MCF7) for gene expression analysis. To assess whether expression of miR-378 and its host gene (PPARGC1B) is downregulated during UPR, HCT116, RKO and MCF7 cells were treated with different EnR stressors: Sarcoplasmic/endoplasmic reticulum calcium ATPase (SERCA) inhibitor thapsigargin (TG), N-linked glycosylation inhibitor Tunicamycin (TM) and Brefeldin A (BFA) inhibitor of anterograde transport from the EnR to the Golgi apparatus. We found that BFA was more potent in reducing cell viability than TG and TM. TG and TM showed cell type-dependent effects on cell viability (Appendix A). We observed more than 50% viability with TG, TM and BFA in all three-cell lines (HCT116, RKO, and MCF7) at the 24 h time point (Appendix A). We observed that treatment of HCT116, RKO, and MCF7 cells with TG, TM and BFA increased the expression of (GRP78, CHOP, and HERP) bonafide UPR-target genes (Figure 3A–C). The increased expression of HERP, GRP78, and CHOP mRNA confirmed the induction of the EnR stress response upon treatment with TG, TM, and BFA. We found that treatment with TG, TM, and BFA decreased the expression of miR-378 and its host gene (PPARGC1B) in all the cell lines tested (Figure 3A–C). However, the fold increase in UPR-target genes and reduction in miR-378 and PPARGC1B in response to EnR stress induced by different pharmacological agents varied quantitatively.

### 2.3. XBP1 Is Required for Downregulation of miR-378

Next, we evaluated the role of three UPR sensors in the downregulation of miR-378 expression. For this, we treated MCF7 cells with BFA, alone or in combination with the IRE1 inhibitor and PERK inhibitor. IRE1 inhibitor efficiently inhibited the BFA-mediated induction of spliced XBP1, and PERK inhibitor compromised the BFA-mediated upregulation of PERK, ATF4, and CHOP mRNA (Figure 4A). These results showed that both PERK and IRE1 inhibitors were efficiently restraining their respective targets. IRE1 inhibitor compromised the BFA-mediated decrease in the expression of miR-378 and PPARGC1B (Figure 4B), whereas PERK inhibitor had no effect on downregulation of miR-378 and PPARGC1B (Figure 4B). To test the role of spliced XBP1 in the downregulation of miR-378, we used the sub-clones of MCF7 cells expressing shRNA targeting XBP1, PERK and ATF6, respectively. As expected, the expression of shRNA reduced the expression of cognate target proteins in the respective sub-clones (Appendix A). We found that knockdown of XBP1 attenuated the BFA-mediated decrease in the expression of miR-378 and PPARGC1B in MCF7 cells, whereas knockdown of ATF6 or PERK had no effect (Figure 4C). Knockdown of XBP1 in HCT116 cells also compromised the decrease in the expression of miR-378 and PPARGC1B during conditions of UPR (Appendix A). To determine whether ectopic XBP1s can regulate miR-378 and PPARGC1B expression, we transfected MCF7 cells with plasmid encoding XBP1s (Figure 4D). We found that ectopic expression of XBP1s resulted in a significant decrease in the expression of miR-378 and PPARGC1B (Figure 4D). In line with the downregulation of PPARGC1B expression by XBP1, we found a negative relationship between the expression of XBP1 transcript levels and mRNA levels of PPARGC1B (significance of correlation: R-value = −0.418, *p*-value = 9.9 × 10^−48^, T-value = −15.245, degrees of freedom = 1095) in breast cancer dataset of TCGA (Figure 4E). Analysis of PPARGC1B promoter sequence using CiiiDER, a tool for predicting and analysing transcription factor binding sites (http://ciiider.org/) [27], identified an XBP1-binding site (Appendix A). Collectively, these results suggest that XBP1s is a negative regulator of miR-378 and PPARGC1B.

### 2.4. miR-378 Reduces Cell Proliferation and Migration of Luminal Cells 

Considering the prognostic value of miR-378 expression in ER-positive breast cancer and TNBC, we generated the sub-clones of MDA-MB-231, MCF7 and ZR-75-1 cells expressing miR-378. HCT116, MDA-MB-231, MCF7, and ZR-75-1 cells were transduced with lentivirus designed to express miRNA-378 and GFP. After confirming the ectopic expression of the miR-378 (Appendix A), we evaluated the effect of miR-378 on cell proliferation and migration, using the sub-clones (miR-CTRL and miR-378) of HCT116, MDA-MB-231, MCF7 and ZR-75-1 cells. We found that miR-378 expression significantly decreased the growth of MCF7 and ZR-75-1 cells; however, it showed no effect on the growth of HCT116 and MDA-MB-231 cells (Figure 5A–D). This observation was further validated by colony formation assay. We observed that miR-378 significantly decreased the number and size of the colonies in MCF7 and ZR-75-1 cells; however, it showed no effect on the colony formation of HCT116 and MDA-MB-231cells (Figure 5A–D).

Considering the effect of miR-378 on cell proliferation in MCF7 and ZR-75-1 cells, we determined the effect of miR-378 on cell migration by in vitro scratch assay. The analysis of bright field images revealed that the rate of wound healing was reduced in miR-378 expressing MCF7 and ZR-75-1 sub-clones as compared to control sub-clones (Figure 6A,B). As per the ATCC (American Type Culture Collection), the population doubling time of MCF7 and ZR-75-1 cells is 29 h and 80 h, respectively. Therefore, the observed differences in the healing of wounds in miR-378 expressing MCF7 and ZR-75-1 cells at early time points were mainly due to migration, whereas at later time points, this effect was the combination of both migration and proliferation. Our results suggest that miR-378 exerts growth-suppressive effects on ER-positive (MCF7 and ZR-75-1) cells.

### 2.5. miR-378 Sensitises Cells to Anti-Estrogens and EnR Stress-Induced Cell Death

We determined whether miR-378 altered the activation of UPR and cell fate during conditions of EnR stress. For this purpose, we examined levels of (CHOP, HERP, GRP78, and XBP1s) UPR target genes to evaluate whether miR-378 modulated their expression. We found that there was no significant difference in the expression of UPR target genes between miR-CTRL and miR-378 cells during UPR conditions (Figure 7A and Appendix A). We found that TG and BFA-induced cell death was enhanced in MCF7-miR-378 cells as compared to MCF7-miR-CTRL cells, while docetaxel (DTX) induced cell death abrogated in MCF7-miR-378 cells (Figure 7B). miR-378 is downregulated in tamoxifen-resistant (TamR) and long-term estrogen-deprived (LTED) MCF7 cells and its downregulation contributes to tamoxifen resistance [26]. Next, we determined the role of miR-378 in response to anti-estrogens. For this, we tested the effect of miR-378 on the sensitivity against (tamoxifen) SERMs and (Fulvestrant) SERDs. We observed that tamoxifen and Fulvestrant treatment significantly reduced the growth of miR-378-expressing MCF7 cells as compared with control cells (Figure 7C). Collectively, these results suggest that miR-378 sensitises MCF7 cells to EnR stress-inducing compounds and anti-estrogens.

### 2.6. miR-378 Dependent Gene Signature Is Associated with Good Outcome

To identify the genes regulated by miR-378, we profiled the differential gene expression induced by miR-378 expression in MCF7 cells. We generated, on average, 43 million clean reads per sample. The gene expression level was quantified by RSEM (RNA-Seq by Expectation Maximization). We found that 53 genes were expressed at a significantly lower level, and 254 were expressed at a significantly higher level in miR-378 expressing cells (Figure 8A,B and Appendix A). To analyse the differentially expressed genes at a functional level, we performed Gene Ontology (GO) enrichment analysis. Gene set enrichment analysis of upregulated genes identified significant enrichment of type I interferon signalling pathway genes, while analysis of downregulated genes showed no significant enrichment (Appendix A). We reasoned that miR-378 expressing cells may secrete growth-suppressing cytokines due to the upregulated type I interferon signalling. Next, we evaluated the effect of miR-378 conditioned-medium on the proliferation of MCF7 cells. We found that the conditioned medium from miR-378 expressing cells reduced the growth of parental MCF7 cells (Figure 8C) as compared to the conditioned medium from control cells. These observations suggest that miR-378 regulates cell growth via the secretion of growth-inhibitory factors. Integration of miR-378-dependent gene expression profiles and GO enrichment analysis identified eight genes (ADRB1, BIRC3, DDX58, LIMCH1, LRRK2, MAOA, SLC16A6, and SP110) upregulated by miR-378. This eight-gene set was designated as the miR-378 signature. Analysis of breast cancer datasets (GSE42568 and GSE2607) using PROGgene V2 demonstrated that patients with an increased miR-378 signature displayed longer overall and relapse-free survival (Figure 8D,E). These findings were validated in another dataset of breast cancer patients using a KM plotter (Figure 8F).

## 3. Discussion

The stressful conditions in the tumour microenvironment, including low oxygen supply, nutrient deprivation, and pH changes, activate a range of cellular stress-response pathways, including activation of the UPR. The miR-378 expression was downregulated in breast cancer samples, and reduced expression of miR-378 was associated with poor outcomes in tamoxifen-treated breast cancer patients [26]. Our data show reduced expression of miR-378 in breast cancer cell lines (Figure 1), and miR-378 expression was significantly associated with better OS in ER-positive breast cancer (Figure 2). Further, we found that patients with an elevated miR-378 signature (eight genes upregulated by miR-378) displayed longer overall and relapse-free survival (Figure 8). Collectively, our data suggest that miR-378 might play an important role in the progression of ER-positive breast cancer. Our results show that XBP1s play a critical role in the downregulation of miR-378 and PPARGC1B during conditions of UPR (Figure 3 and Figure 4, Appendix A). EnR stress and UPR activation contribute to the development and progression of human disease, including neurodegenerative disorders, diabetes, obesity, cancer, and cardiovascular disease [2]. Our results show that the repression of miR-378 by XBP1 (Figure 3 and Figure 4) provides a mechanism for reduced expression of miR-378 in physiological and pathological conditions leading to UPR. Analysis of PPARGC1B promoter sequence using CiiiDER, a tool for predicting and analysing transcription factor binding sites (http://ciiider.org/) [27], identified an XBP1-binding site. This suggests a direct mechanistic link for the suppression of PPARGC1B expression by XBP1s, and further work is required to check if the downregulation of PPARGC1B is mediated by identified XBP1s-binding sites.

While XBP1 is primarily known for its role in EnR stress and protein folding, emerging evidence suggests its involvement in the development and progression of breast cancer. Indeed, several studies have revealed the role of XBP1 in TNBC and ER-positive breast cancers. It has been reported that XBP1s physically interact with Hypoxia Inducible Factor 1α (HIF1α) and c-MYC in the context of TNBC [28]. The crosstalk between XBP1s and estrogen signalling creates a positive feedback loop that results in increased expression of XBP1 in ER-positive breast cancer [23]. XBP1s can contribute to endocrine resistance by enhancing the transcriptional activity of point mutants (Y537S, D538G) and fusion mutants (ESR1-YAP1, ESR1-DAB2) of ESR1 [22]. Knockdown of XBP1 in genome edited MCF7 cells expressing Y537S mutant reduced their growth and re-sensitised them to endocrine therapy [22]. Furthermore, NCOA3 RRM2 and CDC6 are mediators of endocrine resistance downstream of XBP1s in ER-positive breast cancer [20,24]. These studies provide a rationale for combining the XBP1 targeting agents with endocrine therapy to overcome endocrine resistance in breast cancers. Thus, XBP1 can drive tumour progression in TNBC and ER-positive breast cancer via distinct mechanisms due to direct interaction with unique transcription factors and crosstalk with different signalling nodes.

Several studies have indicated that miR-378 is upregulated and acts as OncomiR in cervical cancer [29,30], lung cancer [31,32,33], glioblastoma [34,35,36], and nasopharyngeal carcinoma [37]. However, miR-378 also plays a tumour suppressor role in colon cancer [38,39,40,41], gastric cancer, prostate cancer [42,43], pituitary adenoma [44], medulloblastoma [45], and myelodysplastic syndrome [46]. Our results show that miR-378 expression reduced cell growth and migration of luminal cells (MCF7 and ZR-75-1) but had no such effect on HCT116 and MDA-MB231 cells (Figure 5 and Figure 6). In agreement with our results, inhibition of miR-378 function was shown to increase the migration of MCF7 cells [26]. The miR-378 has been shown to increase migration and invasion without having any effect on the proliferation of MDA-MB231 cells [47]. Further, miR-378 was reported to decrease cell proliferation in HCT116 cells, but we did not observe any effect of miR-378 on the growth of HCT116 cells [40]. This may be due to the differences in the approaches utilised to express the ectopic miR-378. We have used a plasmid-based method for stable expression of miR-378, which will increase the levels of both miR-378 and miR-378*, while Browne et al., and Wang et al., have used transient transfections of miR-378 mimics [40,47]. These observations underscore the context-dependent behaviour of miR-378 in human cancer. The divergent effect of miR-378 in human cancers can be reconciled by since miR-378 has the capacity to target several hundred of different mRNAs, some of which may have opposing oncogenic or tumor-suppressive functions. We propose that oncogenic or tumour-suppressive effect of miR-616 is determined by the relative abundance of oncogenic/tumour suppressor transcripts that can be regulated by miR-378 in a given cellular context.

Our results reveal a pivotal role for miR-378 in determining cell fate during UPR, where ectopic miR-378 sensitised the cells to UPR-induced cell death but provided resistance to docetaxel (Figure 7). Indeed, miR-378 overexpression has been reported to attenuate Aurora B kinase activity and resistance against Taxol [48]. In addition, miR-378 also sensitises the cells to anti-estrogens (Figure 7). The expression of miR-378 has been shown to be downregulated in endocrine-resistant models (TamR and LTED cells) of breast cancer and breast cancer tissues [26]. Knockdown of miR-378 has been shown to reverse the tamoxifen-mediated inhibition of cell growth in MCF7 cells [26]. Considering our results, we speculate that the downregulation of miR-378 by XBP1s may modulate the endocrine resistance in ER-positive breast cancer. Our results indicate that miR-378 is a positive regulator of type I interferon signalling, and miR-378-expressing cells secrete growth-suppressing cytokines (Figure 8C). The studies to elucidate mechanisms by which miR-378 upregulates the type I interferon signalling pathway and identify key growth inhibitory molecules secreted by miR-378-expressing cells are currently underway.

## 4. Materials and Methods

Cell culture and treatments—Human breast cancer cells (MCF7, MDA-MB-231, and ZR-75-1) were purchased from ECACC. Colorectal cancer cells (HCT116 and RKO) were a kind gift from Dr. Victor E. Velculescu, Johns Hopkins University, USA. HEK 293T cells were obtained from Indiana University National Gene Vector Biorepository, USA. The MCF7, MDA-MB-231 and HEK 293T cells were maintained in Dulbecco’s modified eagle’s medium (DMEM) (Sigma, London, UK, Cat#D6429). HCT116 and RKO cells were maintained in McCoy’s 5A modified medium (Sigma, London, UK, Cat#M9309), and ZR-75-1 cells were maintained in RPMI-1640 (Sigma, London, UK, Cat#R4130), and all media contained 10% heat-inactivated fetal bovine serum (FBS) and 100 U/mL penicillin and 100 μg/mL streptomycin (Sigma, London, UK, Cat#P0781) with 5% CO_2_ at 37 °C.

Thapsigargin (Cat#1138), Tunicamycin (Cat# 3516), BFA (Cat#1231), PERK inhibitor GSK2606414 (Millipore Ireland, Cat#516535) and IRE1 Inhibitor (4μ8C) (Millipore Ireland, Cat#412512) were purchased from Tocris Bioscience. Fulvestrant (Sigma, London, UK, Cat#I4409), tamoxifen (Sigma, London, UK, Cat#T5648) and Docetaxel (Sigma, London, UK) were used for cell death assay at the optimised concentrations for the indicated time points. All reagents were purchased from Sigma–Aldrich unless otherwise stated.

Plasmid constructs—The PERK shRNA plasmid, ATF6 shRNA plasmid and XBP1 shRNA plasmid have been described previously [20,49]. miExpress^TM^ precursor miRNA expression clones, miR-CTRL and miR-378 (pEZX-MR03 vector) were sourced from GeneCopoeia, Rockville, MD, USA.

Generation of stable sub-clones—Lentiviruses were generated by transfecting lentiviral transfer plasmids along with packaging plasmids in 293T cells, as described previously [50]. Lentivirus containing supernatant was used to generate stable miR-378 overexpressing sub-clones of HCT116, MCF7, ZR75-1, and MDA-MB231 cells and knockdown sub-clones of HCT116 and MCF7 cells for UPR genes (PERK, XBP1, and ATF6). After transduction, cells were grown in the media with 1 µg/mL puromycin for 7 days. The percentage of miR-378 transduced cells was evaluated based on GFP expression under a fluorescence microscope.

RNA extraction, Reverse transcription reaction and real-time quantitative PCR—Total cellular RNA was purified using Trizol reagent (Life Technologies, Fischer Scientific, Dublin, Ireland). Reverse transcription (RT) was carried out using ImProm-II™ Reverse Transcription System (Promega, MyBio, Kilkenny, Ireland) per the manufacturer’s instructions. Real-time PCR procedure to determine the expression of UPR-regulated genes. The sequence of primers and probes used are listed in Appendix A. GAPDH (for mRNAs) and RNU6B (for miRNAs) were used as reference genes to determine the relative expression level of target genes between treated and control samples using the ΔΔCt method.

Colony formation assay—The control and miR-378 expressing cells (1000 cells per well) were plated in 6-well plates and were grown for 14 days. Then, cells were washed twice with PBS and fixed with 10% formaldehyde for 5 min and stained with 0.5% crystal violet for 10 min. The number of colonies that had more than 50 cells was counted in five random view fields under a microscope, and the average number of colonies was achieved. Colony size was also determined using Image J software.

MTS cell proliferation assay—The control and miR-378 expressing cells (2 × 10^3^ cells/well) were plated in 96 well plates. After 24 h of growth, for cytotoxicity assay, cells were treated with (100 µM) Fulvestrant (Sigma, UK, Cat#I4409) and (10 µM) tamoxifen for 0, 24, 48 and 72 h. The 0.02 mL of the MTS solution (Promega, Cat#G3582) was directly added into each well and incubated at 37 °C for 4 h. The absorbance of each well was measured at OD = 490 nm with a 96-well plate reader (BIO-TEK Synergy plate reader, Agilent Technologies Ireland Limited, Cork, Ireland).

Scratch wound healing assay—The control and miR-378 expressing cells (3 × 10^5^ cells/well) were plated in 6-well plates. After 24 h of growth, when they reached 70–80% confluency, the cell monolayer was scratched with a sterile 0.2 mL pipette tip across the centre of the well. Each well was then washed twice with medium to remove the detached cells and then replaced with fresh medium. The scratch areas were imaged at different time points, including 0 (after creating the scratch), 12, 24 and 48 h and the area of the scratch was quantified by Image J software.

Western blot analysis—Whole cell lysates were analysed by western blotting. The nitrocellulose membranes were blocked with the specific blocking solution for 2 h at room temperature. The nitrocellulose membranes were then treated with specific primary antibodies, including PERK (Cell Signalling, Brennan & Co, Stillorgan, Dublin Ireland, Cat#3192), ATF6 (Abcam, Abcam (Netherlands BV), Amsterdam, The Netherlands, Cat#ab122897), Spliced XBP1 (Bio Legend, San Diego, CA, USA, Cat#619501) and β-Actin (Sigma, Cat#A-5060) at 4 °C overnight. The dilutions of antibodies used are described in Appendix A. After washing three times with PBS/0.05%Tween solution, the membranes were incubated with appropriate horseradish peroxidase-conjugated secondary antibody at room temperature for 2 h. The membranes were then washed twice with PBS/0.05%Tween and once with PBS, and finally, the signals were detected using Western Lightening chemiluminescent substrate (Perkin Elmer, Groningen, The Netherlands, Cat#NEL104001EA).

Flow cytometry analysis—The control and miR-378 expressing MCF7 cells (2 × 10^5^ cells/well) were plated in 6 well plates. After 24 h of plating, cells were either untreated or treated with Thapsigargin (0.1 µM), Tunicamycin (0.05 µg/mL), Brefeldin A (0.05 µg/mL) and Docetaxel (100 nM) for 48 h and analysed by flow cytometry [49]. Briefly, cells were then harvested and washed with ice-cold FACS Buffer (PBS, 0.5–1% BSA or 5–10% FBS and 0.1% NaN3 sodium azide). Cell concentration was adjusted to 1 × 10^6^ cells/mL, and 1 μL of SYTOX Red dye (Invitrogen, UK, Cat#S34859) was added to each flow cytometry tube. Cells were incubated for 15 min at 4 °C, protected from light. Then, cells were analysed by a flow cytometry machine (BD Accuri™ C6 Plus, BD Biosciences, Berkshire, UK) using 635 nm for excitation and 660 nm for emission.

Analysis of Oncomine database—The oncomine cancer microarray database (Compendia Biosciences; Ann Arbor, MI, USA) [51] was used to examine the expression of PPARGC1B in cancer tissue (GSE2109).

High-throughput mRNA sequencing and analysis—RNA-seq was used to determine the mRNA expression profiles in control and miR-378 expressing MCF7 cells (four replicates) by BGI Tech Solutions, Hong Kong. We obtained an average of 43,802,853 raw seq53uencing reads and 43,749,166 clean reads (after filtering low-quality reads) for each sample. Clean reads were then mapped to reference using HISAT and Bowtie2. The average mapping ratio with reference gene was 82.81%, and the average genome mapping ratio was 95.95%. FPKM method was used to calculate the expression level, using the following formula: FPKM = 10^6^C/(NL/10^3^).

For the expression of a given gene A, C is the number of fragments that are aligned to gene A, N is the total number of fragments that are aligned to all genes, and L is the number of bases present in gene A. We used the NOISeq method to screen for differentially expressed genes between control and miR-378 expressing groups. NOISeq uses the sample’s gene expression in each group to calculate log2fold change (M) and absolute different value (D) of the control and miR-378 expression group to build a noise distribution model. For any given gene A, NOISeq computes its average expression “Control_avg” in the control group and average expression “Treat_avg” in the treatment group. Next, the fold change and absolute different value were analysed using the following formula:Fold change for gene A, MA = log2 {(Treat_avg)/(Control_avg)}
Absolute different value for gene A, DA = |Control_avg-Treat_avg|

If MA and DA diverge markedly from the noise distribution model, gene A will be defined as a differentially expressed gene (DEG). There is a probability value to assess how MA and DA both diverge from the noise distribution model. Finally, DEG was identified according to the following default criteria: Fold change ≥ 2 and diverge probability ≥ 0.8.

Preparation of conditioned media—Conditioned media derived from control and miR-378 expressing MCF7 cells were prepared as follows: 1 × 10^6^ cells were cultured on a T25 flask with 5 mL of complete DMEM for 12 h. The medium was harvested and filter-sterilised using a 0.22-μm Millex-HV syringe filter (Merck Millipore, Arklow, Co. Wicklow Ireland).

Survival analysis for miR-378—Kaplan–Meier Plotter (http://kmplot.com/analysis/), a database that integrates gene expression data and clinical data, was used to determine the prognostic value of miR-378a in breast cancer. We focused the analysis on subtypes of breast cancer and determined overall survival using the TCGA dataset.

The patient samples were separated into two groups based on miR-378a expression. The hazard ratio with 95% confidence intervals and log-rank *p*-value was calculated.

Survival analysis for miR-378 signature—The survival analysis was performed using the gene expression profiles of 1809 breast cancer samples available at Kaplan–Meier Plotter (http://kmplot.com/analysis/) and GSE42568 and GSE2607 breast cancer datasets using PROGgene V2 prognostic database (http://watson.compbio.iupui.edu/chirayu/proggene/database/index.php, accessed on 8 June, 2018). We divided patients into two subgroups based on the expression of miR-378 gene signature. Patients who had an average expression of all the genes in the miR-378 signature, higher than the median, were named as “high miR-378 signature”. Patients who had an average expression of all the genes in the miR-378 signature that was lower than the median were named “low miR-378 signature”. Kaplan–Meier survival analysis was used to evaluate the statistical significance of the outcome between the two groups.

Statistical Analysis—The data were analysed using the software package SPSS 21.0 for Windows, and a two-tailed unpaired *t*-test was performed to determine any statistically significant differences between independent groups. Results with a *p* < 0.05 were considered statistically significant. All experiments were performed in triplicates.

## 5. Conclusions

We show that XBP1s downregulate the expression of PPARGC1B/miR-378 during conditions of UPR. The ectopic expression of miR-378 significantly suppresses cell growth, proliferation, and migration and sensitises cells to UPR-induced cell death and anti-estrogens. Higher expression of miR-378 was associated with better overall survival in ER-positive breast cancer. We found that miR-378-expressing cells secrete growth-suppressive cytokines, which is both necessary and sufficient for miR-378-mediated growth inhibition (Figure 9). Collectively, our findings reveal a crucial role for the spliced XBP1-miR-378 pathway in ER-positive breast cancer and suggest that XBP1s may contribute to the development of endocrine-resistant breast cancer, in part, by downregulating the expression of miR-378. The targeting of XBP1 has been considered a promising therapeutic approach to overcome endocrine resistance in breast cancer. The efficacy of ORIN1001 (IRE1 inhibitor that blocks XBP1s production) is being evaluated in a phase 1 trial in patients with advanced solid tumours or relapsed refractory metastatic breast cancer (NCT03950570). It will be interesting to evaluate the modulation of the XBP1s-miR-378 axis in patients treated with ORIN1001.

## Figures and Tables

**Figure 1 ijms-25-00186-f001:**
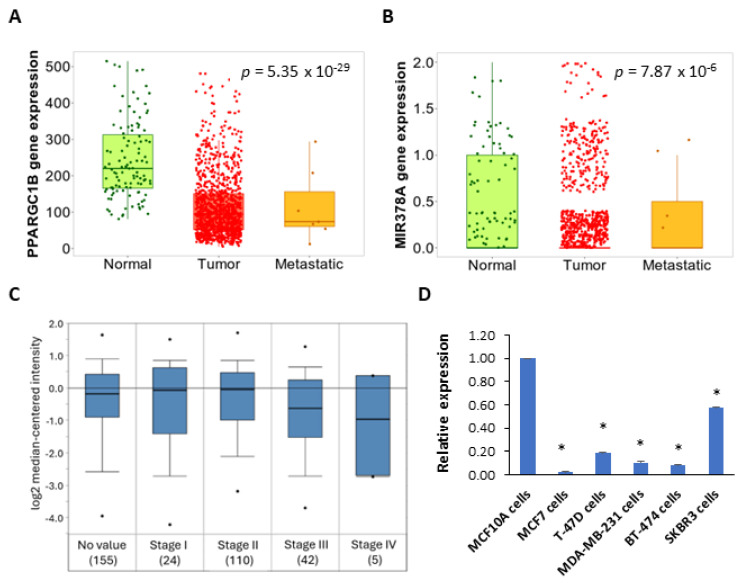
Expression of miR-378 and PPARGC1B is downregulated in breast cancer. TNMplot (https://tnmplot.com/analysis/ accessed on 12 September 2023) was used to determine the expression of PPARGC1B and miR-378 in normal (*n* = 113), tumour (*n* = 1097) and metastatic (7) tissues for human breast cancers. A box plot for expression of (**A**) PPARGC1B and (**B**) miR-378 is shown. (**C**) Box plots showing the expression of the PPARGC1B gene in the indicated categories of ductal breast carcinoma as derived from the Oncomine database. (**D**) Expression level of miR-378 as quantified by RT−qPCR, normalising against RNU6b. Error bars represent mean ± S.D. from three independent experiments performed in triplicate. * *p* < 0.05, two-tailed unpaired *t*-test as compared to the untreated control.

**Figure 2 ijms-25-00186-f002:**
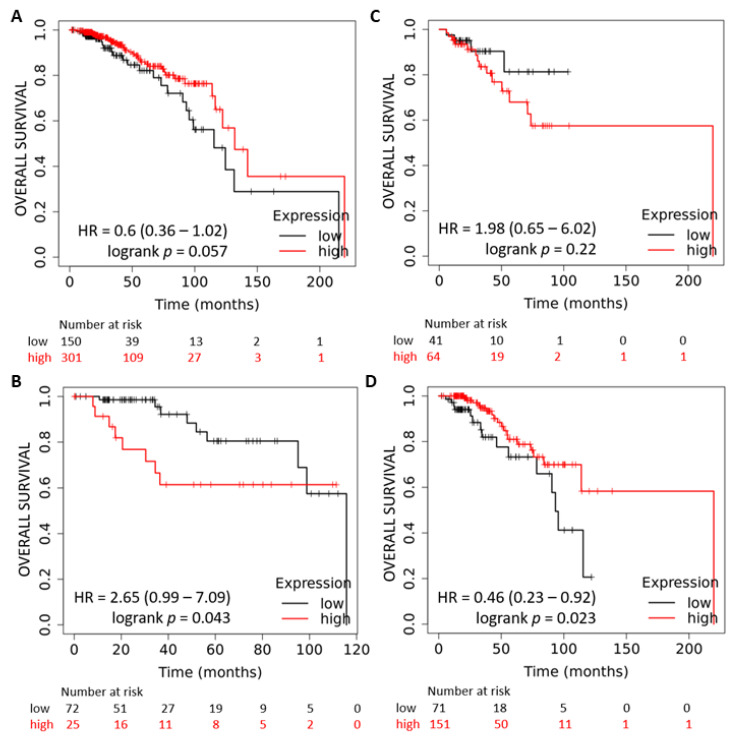
High expression of miR-378 is associated with better overall survival in ER-positive breast cancer. Kaplan–Meier plots showing overall survival in subtypes of breast cancer divided based on miR-378a expression from the TCGA dataset. (**A**) ER-positive patients (*n* = 451); (**B**) TNBC patients (*n* = 97); (**C**) HER2-enriched breast cancer (*n* = 105); and (**D**) ER-positive and node-positive patients (*n* = 222). In red, patients with expressions above the cut-off are used in the analysis, and in black, patients with expressions below the cut-off are used in the analysis.

**Figure 3 ijms-25-00186-f003:**
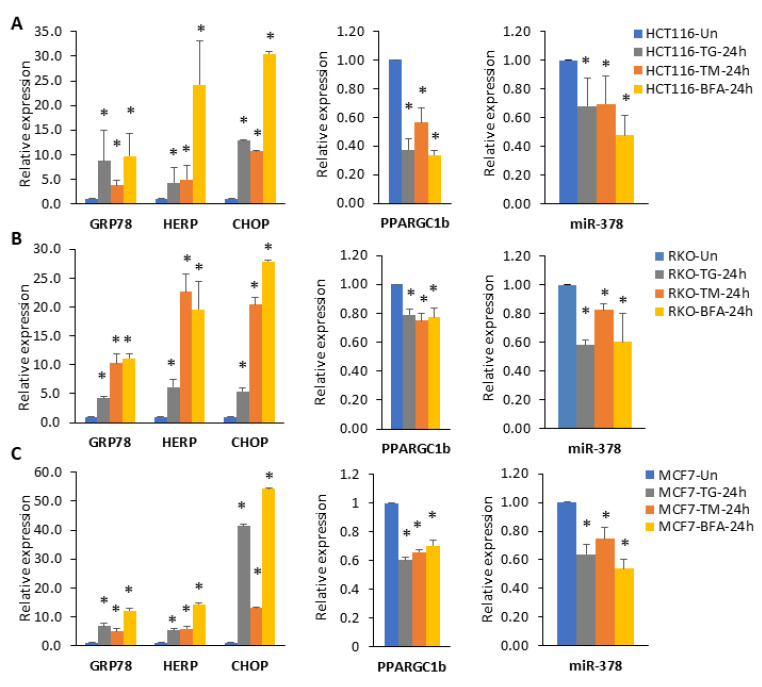
Downregulation of miR-378 and its host gene (PPARGC1B) during conditions of UPR. (**A**) HCT116 cells, (**B**) RKO cells and (**C**) MCF7 cells were either untreated or treated with (1 µM) TG, (1.0 μg/mL) TM and (0.5 μg/mL) BFA for 24 h. The change in expression levels of UPR (GRP78, HERP, and CHOP) and PPARGC1B was quantified by RT–qPCR, normalising against GAPDH. The expression levels of miR-378 were quantified by RT–qPCR, normalising against RNU6b. Error bars represent mean ± S.D. from three independent experiments performed in triplicate. * *p* < 0.05, two-tailed unpaired *t*-test as compared to untreated control.

**Figure 4 ijms-25-00186-f004:**
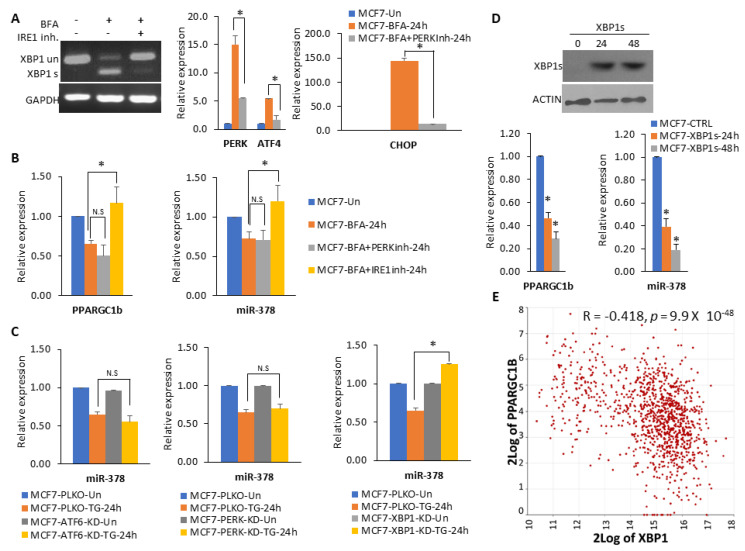
Downregulation of miR-378 during EnR stress is dependent on the IRE1–XBP1 axis. (**A**) MCF7 cells were either untreated or treated with (0.5 μg/mL) BFA in the absence and presence of IRE1 inhibitor, (10 µM) 4µ8C and GSK PERK inhibitor (100 nM) for 24 h. The expression of XBP1 (unspliced and spliced), GAPDH, PERK, ATF4, and CHOP was analysed by RT–qPCR. (**B**) Cells were treated as in A; GAPDH normalised levels of PPARGC1B and RNU6b normalised levels of miR-378 are shown. (**C**) MCF7-PKLO, MCF7-XBP1-KD, MCF7-PERK-KD, and MCF7-ATF6-KD were treated with (0.5 μg/mL) BFA for 24 h. The expression level of miR-378 was quantified by RT-qPCR, normalising against RNU6b. (**D**) MCF7 cells were transfected with a plasmid expressing spliced XBP1. Cells were harvested at indicated time points, and immunoblotting was performed using antibodies against spliced XBP1 and β-actin. The change in expression levels of PPARGC1B and miR-378 was quantified by RT–qPCR. (**E**) The dot plot of log2 transformed values for co-expression of PPARGC1B and XBP1 as determined by R2 Genomics Analysis and Visualization Platform is shown. Error bars represent mean ± S.D. from three independent experiments performed in triplicate. * *p* < 0.05, two-tailed unpaired *t*-test as compared to untreated control. N.S., not significant.

**Figure 5 ijms-25-00186-f005:**
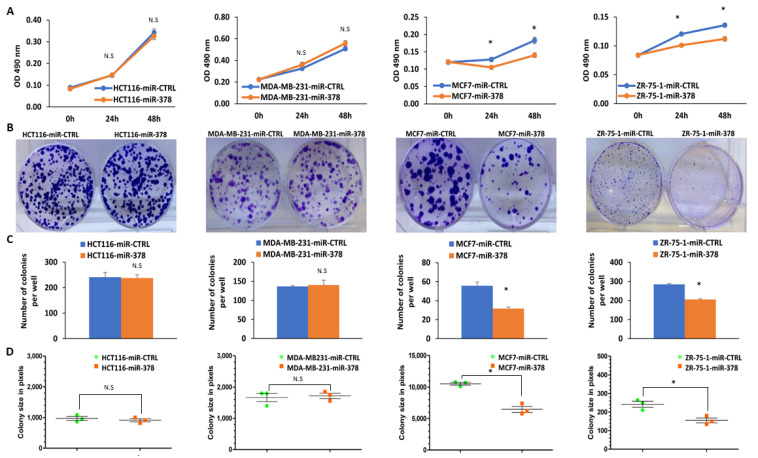
The ectopic miR-378 reduces the proliferation of ER-positive breast cancer cells. (**A**) Control and miR-378 expressing (HCT116, MDA-MB-231, MCF7, and ZR-75-1) sub-clones were plated in 96-well plates (2000 cells/well). Cell proliferation was assessed by MTS assay. Line graphs show the absorbance in cells at the indicated time points. Error bars represent mean ± S.D. from three independent experiments performed in triplicate. (**B**–**D**) Control and miR-378 expressing (HCT116, MDA-MB-231, MCF7 and ZR-75-1) sub-clones were plated in 6-well plates (1000 cells/well) and grown for 14 days. (**B**) Colonies stained with crystal violet are shown. The lower panels show the quantification of colony (**C**) number and (**D**) size per well for control and miR-378 expressing sub-clones. Error bars represent mean ± S.D. from three independent experiments performed in triplicate. * *p* < 0.05. N.S., not significant.

**Figure 6 ijms-25-00186-f006:**
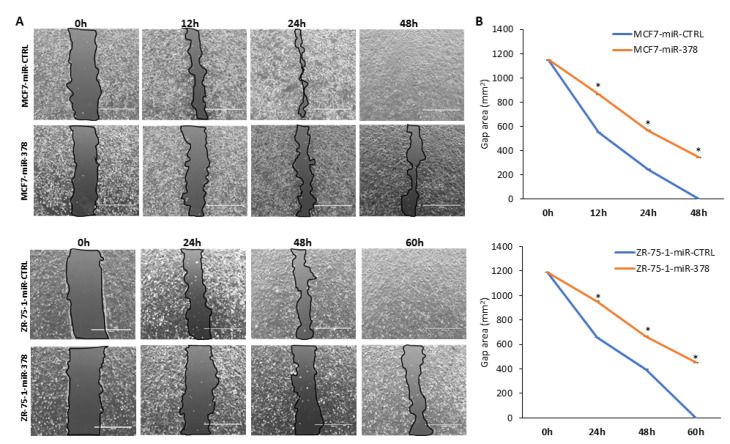
MiR-378 reduces the migration of ER-positive breast cancer cells. (**A**) Control and miR-378 expressing (MCF7 and ZR-75-1) sub-clones were plated in a 6-well plate. The cells were scratch wounded with a micropipette tip (200 µL). Black lines indicate the wound borders at the beginning of the assay and were recorded at indicated time points post-scratching. Scale bar, 1000 µM. (**B**) Line graph shows the scratch gap quantified using Image J software Version 1.51 at the indicated time points. Error bars represent mean ± S.D. from three independent experiments. * *p* < 0.05.

**Figure 7 ijms-25-00186-f007:**
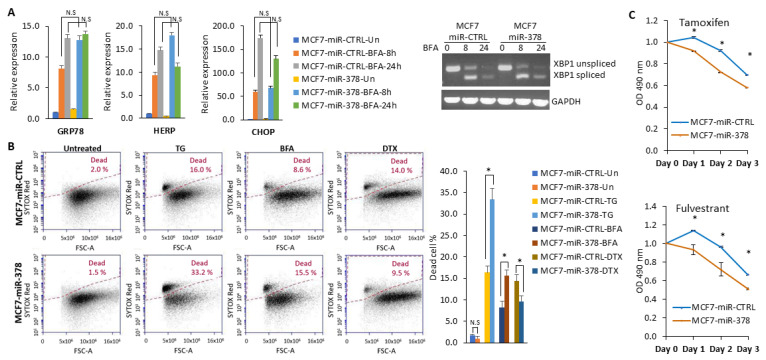
The miR-378 enhances the sensitivity towards EnR stress-inducing compounds. (**A**) Control and miR-378 expressing MCF7 sub-clones were either untreated or treated with (0.5 μg/mL) BFA for 8 and 24 h. The change in expression levels of UPR target genes (GRP78, HERP, CHOP, and XBP1) were quantified by RT-qPCR, normalising against GAPDH. Error bars represent mean ± S.D. from three independent experiments performed in triplicate. (**B**) Control and miR-378 expressing MCF7 sub-clones were either untreated or treated with (1 µM) TG, (0.5 μg/mL) BFA and (100 nM) DTX for 48 h. A representative dot plot of SYTOX Red staining is shown. Bar graphs show the percentage of SYTOX-positive cells. Error bars represent mean ± S.D. from two independent experiments. * *p* < 0.05, two-tailed unpaired *t*-test as compared to untreated control. (**C**) Control and miR-378 expressing MCF7 cells were plated in 96-well plates (2000 cells/well) and then either untreated or treated with (10 µM) tamoxifen and (100 µM) Fulvestrant. Cell proliferation was assessed by MTS assay at different time intervals (0, 24, 48 and 72 h). O.D 490 at 0 h was arbitrarily set at one, and relative change in relative absorbance at subsequent time points is shown. Error bars represent mean ± S.D. from three independent experiments performed in triplicate. N.S., not significant.

**Figure 8 ijms-25-00186-f008:**
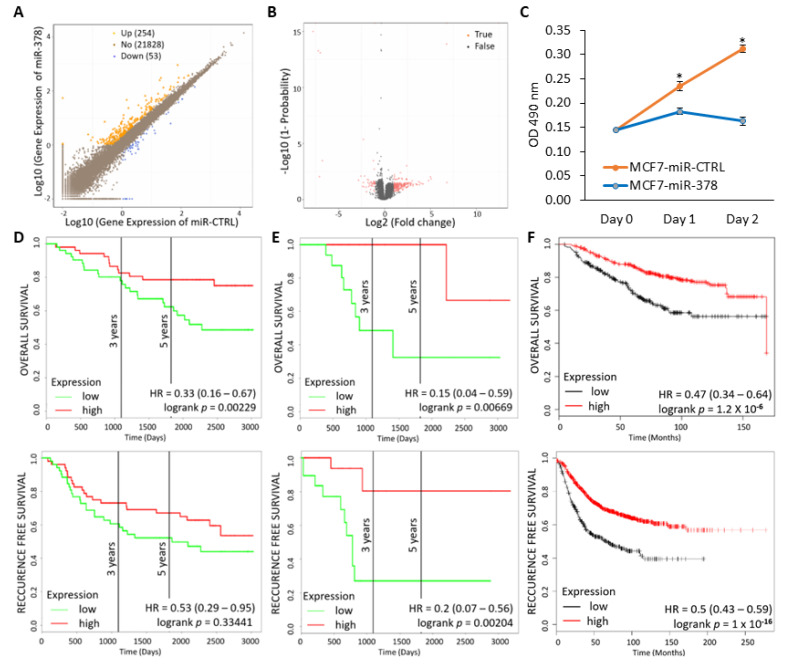
miR-378 gene signature is associated with prognosis in breast cancer. (**A**) Scatter plot of expressed genes in control and miR-378 expressing MCF7 cells as determined by NOISeq method. The *X*-axis and *Y*-axis present the log2 value of gene expression in control and miR-378 expressing cells. (**B**) Volcano plot of differentially expressed genes in control and miR-378 expressing MCF7 cells. The *X*-axis represents the fold change, and the *Y*-axis represents the threshold value. Each dot is a differential expressed gene. Dots in red mean significantly differential expressed genes that passed the screening threshold, and black dots are non-significant differential expressed genes. (**C**) MCF7 were incubated with the control or miR-378 conditioned medium in a 96-well plate (2000 cells/well). Cell proliferation was assessed by MTS assay. Line graphs show the absorbance in cells at the indicated time points. Error bars represent mean ± S.D. from two independent experiments performed in triplicate. (**D**,**E**) Kaplan–Meier graphs demonstrate a significant association of elevated expression of the miR-378 gene signature (red line) with increased overall survival and recurrence-free survival in two cohorts of breast cancer patients using PROGgene V2. (**F**) Kaplan–Meier graphs demonstrate a significant association between elevated expression of the miR-378 signature (red line) and longer relapse-free survival in 1809 patients using the KM plotter. * *p* < 0.05.

**Figure 9 ijms-25-00186-f009:**
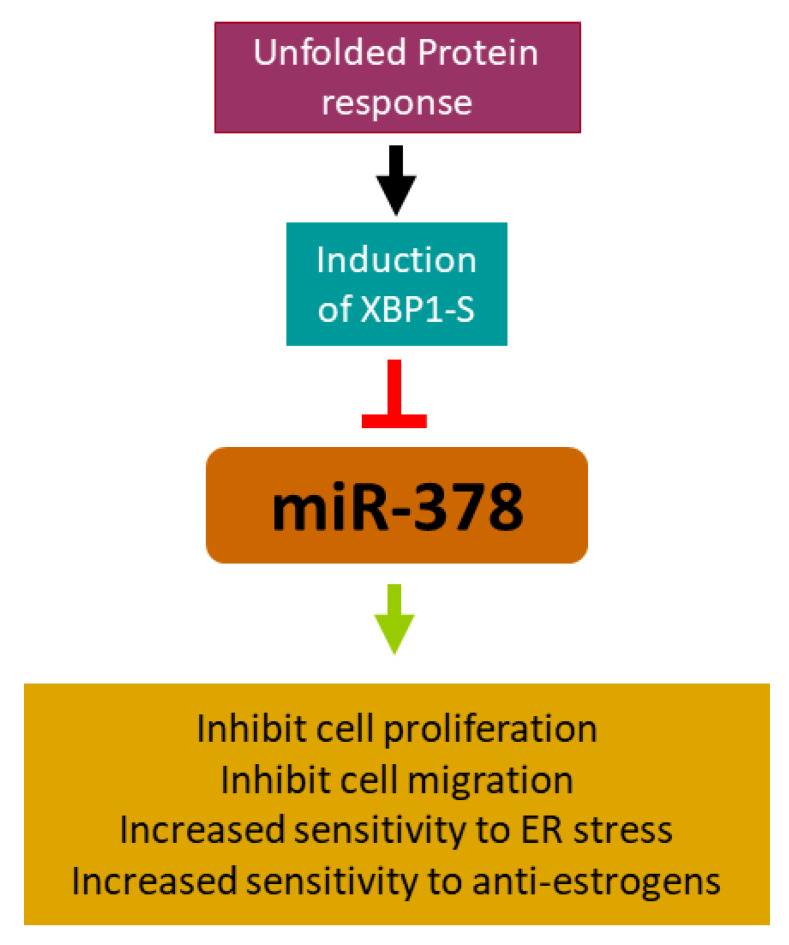
Stressful conditions of the tumour microenvironment activate an adaptive mechanism called the unfolded protein response (UPR). X-box binding protein-1 (XBP1), a critical transcriptional activator, is induced by the UPR and downregulates the expression of miR-378 during conditions of UPR. Ectopic expression of miR-378 reduces the growth and migration of ER-positive breast cancer cells and sensitises ER-positive breast cancer cells to anti-estrogens. XBP1s may contribute to the development of endocrine-resistant breast cancer, in part by downregulating the expression of miR-378.

## Data Availability

The datasets supporting the conclusions of this article are included within the article. All other datasets and materials used during the current study are available from the corresponding author upon reasonable request.

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
