# Peer review of "miRNA-378 Is Downregulated by XBP1 and Inhibits Growth and Migration of Luminal Breast Cancer Cells"

_ijms, 2023, doi:10.3390/ijms25010186_

Round 1

Reviewer 1 Report

Comments and Suggestions for Authors

Comments for authors:

Title:

The title accurately reflects the content of the study and is concise and informative.

Introduction:

1.      The term "misfolded ERHT" is introduced without prior explanation, and it might benefit the reader to briefly define or explain this term to maintain clarity..

2.      The section mentions that E2-stimulation induces significant protein synthesis and activates UPR, but it could be enhanced by briefly explaining why UPR activation is required for E2-mediated cell proliferation in breast cancer cells.

3.      While the study discusses the role of XBP1 in endocrine resistance, it would be beneficial to integrate these findings more explicitly into the broader context of breast cancer biology and treatment, providing a seamless narrative for the reader.

4.      Expression Regulation of miR-378: The section mentions that not much is known about the mechanisms regulating the expression of miR-378. Providing a brief overview or hypotheses on potential regulators could enhance the completeness of the discussion.

Methods:

1.      Could include information on passages used for experiments, ensuring consistency.

2.      It would be beneficial to explain the rationale behind selecting specific concentrations for treatments

3.      The efficiency of transduction and stability of sub-clones could be discussed.

4.      The biological relevance of the selected functional assays could be discussed.

5.      Validation of antibody specificity and controls for flow cytometry could be addressed.

Results:

1.      The association of miR-378 expression with overall survival (OS) in ER-positive breast cancer is shown through Kaplan-Meier plots, providing evidence for its prognostic value supporting the research hypothesis

2.      The role of the IRE1-XBP1 axis in the downregulation of miR-378 is investigated through pharmacological inhibition and shRNA knockdown experiments, adding credibility to the proposed mechanism

3.      The study investigates the functional impact of miR-378 by assessing its effect on cell proliferation, colony formation, and migration in ER-positive breast cancer cells. The findings suggest a growth-suppressive role for miR-378. Please emphasize this

Discussion:

1.      While the integration of previous research is well-done, further details on the specific studies, methodologies, and key findings could enhance the section.

2.      The section could benefit from a brief explanation of why ER-positive breast cancer was specifically chosen for investigation.

3.      Please Acknowledges the context-dependent behavior of miR-378 in different human cancers.

4.      A brief discussion on the potential clinical significance of miR-378's role in type I interferon signaling could enhance this part.

Conclusions

1.      A brief mention of the potential clinical implications or significance of targeting the XBP1s-miR-378 pathway in ER-positive breast cancer could add depth.

Author Response

Reviewer 1

Title:

The title accurately reflects the content of the study and is concise and informative.

Introduction:

  1. The term "misfolded ERHT" is introduced without prior explanation, and it might benefit the reader to briefly define or explain this term to maintain clarity..

Author response: We thank the reviewer for this suggestion. The following text is added to the revised manuscript.

Labelling HaloTag proteins with low molecular weight hydrophobic tags (HyTs) leads to targeted degradation of HaloTag fusion proteins. HyT-induced destabilization of EnR-localized HaloTag (ERHT) protein robustly activates transient and non-apoptotic UPR, accompanied by gene expression signature reflecting estrogen signalling [13].  Inhibition of the IRE1-XBP1 axis abrogated the activation of the estrogen pathway by HyT-induced destabilization of ERHT [13]. Page 3 of revised manuscript

  1. The section mentions that E2-stimulation induces significant protein synthesis and activates UPR, but it could be enhanced by briefly explaining why UPR activation is required for E2-mediated cell proliferation in breast cancer cells.

Author response: We thank the reviewer for this suggestion. The following text is added to the revised manuscript.

UPR induced by E2-ERα increases the protein folding capacity of EnR and thereby primes cells to meet the higher protein handling demands required for the later growth phases following E2 stimulation. Page 3 of revised manuscript

  1. While the study discusses the role of XBP1 in endocrine resistance, it would be beneficial to integrate these findings more explicitly into the broader context of breast cancer biology and treatment, providing a seamless narrative for the reader.

Author response: We thank the reviewer for this suggestion. The following text is added to the revised manuscript.

Estrogen signaling transiently activates all three sensors of the UPR, but activation of the IRE1-XBP1 axis is preferentially sustained by a combination of non-genomic and genomic actions of estrogen, thereby generating a positive feedback regulatory loop consisting of spliced XBP1 and ER. Page 3-4 of revised manuscript

  1. Expression Regulation of miR-378: The section mentions that not much is known about the mechanisms regulating the expression of miR-378. Providing a brief overview or hypotheses on potential regulators could enhance the completeness of the discussion.

Author response: We thank the reviewer for this suggestion. The following text is added to the revised manuscript.

The expression of miR-378 and its host gene, PPARGC1B is downregulated in tumour samples from breast cancer patients. We hypothesize that induction of UPR in tumour microenvironment may play a role in reduced expression of miR-378 in breast cancer. Page 4 of revised manuscript

Methods:

  1. Could include information on passages used for experiments, ensuring consistency.

 Author response: After revival from liquid nitrogen the sub-clones were maintained for 12-14 passages for the experiments. After 2-3 months in culture cells were discarded, and a new batch was revived.

  1. It would be beneficial to explain the rationale behind selecting specific concentrations for treatments.

Author response: We thank the reviewer for this suggestion. The following text is added to the revised manuscript.

The activation kinetics of three UPR sensors in response to stress induced by different pharmacological agents has shown fundamental differences in their recognition of different forms of EnR stress. Since microarray analysis was performed in, colorectal cells we used cell lines from colorectal (HCT116 and RKO) and breast cancer (MCF7) for gene expression analysis. To assess whether expression of miR-378 and its host gene (PPARGC1B) is downregulated during UPR, HCT116, RKO and MCF7 cells were treated with different EnR stressors: Sarcoplasmic/endoplasmic reticulum calcium ATPase (SERCA) inhibitor thapsigargin (TG), N-linked glycosylation inhibitor Tunicamycin (TM) and Brefeldin A (BFA) inhibitor of anterograde transport from the EnR to the Golgi apparatus [32]. We found that BFA was more potent in reducing cell viability as compared to TG and TM. TG and TM showed cell type-dependent effects on cell viability (SF1). We observed more than 50% viability with TG, TM and BFA in all three-cell lines (HCT116, RKO and MCF7) at 24h time point (SF 1). Page 11-12 of revised manuscript

  1. The efficiency of transduction and stability of sub-clones could be discussed.

Author response: We did not determine the efficiency of transduction per se. To generate the sub-clones, after transduction cells were selected with 1 µg/ml puromycin for 7 days. The stability of sub-clones was evaluated by monitoring GFP expression and qRT-PCR for miR-378. We did not notice any change in percentage of GFP positive cells and expression of miR-378 over time.

  1. The biological relevance of the selected functional assays could be discussed.

Author response: All the functional assays used were appropriate to answer the question.

  1. Validation of antibody specificity and controls for flow cytometry could be addressed.

Author response: We are confident about the specificity of ATF6, PERK and XBP1 antibody used for analysis because we see clear blots with bands in the expected size range which were reduced in the corresponding knock-down sub-clones.

Results:

  1. The association of miR-378 expression with overall survival (OS) in ER-positive breast cancer is shown through Kaplan-Meier plots, providing evidence for its prognostic value supporting the research hypothesis
  2. The role of the IRE1-XBP1 axis in the downregulation of miR-378 is investigated through pharmacological inhibition and shRNA knockdown experiments, adding credibility to the proposed mechanism
  3. The study investigates the functional impact of miR-378 by assessing its effect on cell proliferation, colony formation, and migration in ER-positive breast cancer cells. The findings suggest a growth-suppressive role for miR-378. Please emphasize this

Author response: We thank the reviewer for the appreciation and encouraging comments. The growth-suppressive role for miR-378 in ER-positive breast cancer is emphasized in the revised abstract.

Discussion:

  1. While the integration of previous research is well-done, further details on the specific studies, methodologies, and key findings could enhance the section.

 Author’s response: We thank the reviewer for the suggestion. We have made the suggested changes in the discussion of revised manuscript.

  1. The section could benefit from a brief explanation of why ER-positive breast cancer was specifically chosen for investigation.

Author response: We thank the reviewer for this suggestion. The following text is added to the revised manuscript.

The expression of miR-378 has been shown to be downregulated in endocrine resistant models (TamR and LTED cells) of breast cancer and breast cancer tissues [31]. Knockdown of miR-378 has been shown to reverse the tamoxifen-mediated inhibition of cell growth in MCF7 cells [31]. Considering our results, we speculate that downregulation of miR-378 by XBP1s may modulate the endocrine resistance in ER-positive breast cancer. Page 20 of revised manuscript

The miR-378 expression was downregulated in breast cancer samples and reduced expression of miR-378 was associated with poor outcome in tamoxifen treated breast cancer patients [31].  Page 18 of revised manuscript

  1. Please Acknowledges the context-dependent behaviour of miR-378 in different human cancers.

Author response: We thank the reviewer for this suggestion. The following text is added to the revised manuscript.

Our results show that miR-378 expression reduced cell growth and migration of luminal cells (MCF7 and ZR-75-1) but had no such effect on HCT116 and MDA-MB231 cells (Fig 5-6). Page 19 of revised manuscript

The divergent effect of miR-378 in human cancers can be reconciled by considering the fact that miR-378 has the capacity to target several hundred of different mRNAs, some of which may have opposing oncogenic or tumor-suppressive functions. We propose that oncogenic or tumour-suppressive effect of miR-616 is determined by the relative abundance of oncogenic/tumour suppressor transcripts that can be regulated by miR-378 in a given cellular context. Page 20 of revised manuscript

  1. A brief discussion on the potential clinical significance of miR-378's role in type I interferon signaling could enhance this part.

Author response: We thank the reviewer for this suggestion. The following text is added to the revised manuscript.

Interestingly, in IFN-a activated natural killer (NK) cells the expression of miRNA-378 was downregulated where it contributed to the cytotoxicity of human NK cells by targeting cytolytic molecules granzyme B and perforin [58]. PPARGC1B has been shown to be a downstream target of IFN-g and it acts in conjunction with ERRa, as a key effector of IFN-g induced mitochondrial ROS production and host defence [59]. Page 20 of revised manuscript

Conclusions

  1. A brief mention of the potential clinical implications or significance of targeting the XBP1s-miR-378 pathway in ER-positive breast cancer could add depth.

Author response: We thank the reviewer for this suggestion. The following text is added to the revised manuscript.

The targeting of XBP1 has been considered a promising therapeutic approach to overcome endocrine resistance in breast cancer. The efficacy of ORIN1001 (IRE1 inhibitor that blocks XBP1s production) is being evaluated in phase 1 trial in patients with advanced solid tumours or relapsed refractory metastatic breast cancer (NCT03950570). It will be interesting to evaluate the modulation of XBP1s-miR-378 axis in patients treated with ORIN1001. Page 21 of revised manuscript

Reviewer 2 Report

Comments and Suggestions for Authors

Review comment

The manuscript titled as "miRNA-378 Is Downregulated by XBP1 and Inhibits Growth and Migration of Luminal Breast Cancer Cells" provides a comprehensive analysis of the role of miRNA-378 in breast cancer, particularly in the context of endoplasmic reticulum (ER) stress and endocrine resistance. Authors concluded that miR-378 upregulates the expression of several genes that regulate type I interferon signalling. Analysis of separate cohorts of breast cancer patients showed that a gene signature derived from miR-378 upregulated genes showed a strong association with improved overall and recurrence free survival in breast cancer. Our results suggest that downregulation of miR-378 by XBP1 contributes to endocrine resistance in ER-positive breast cancer. This study possesses certain novelty, but the evidence should be further enhanced. Thus, major revision is recommended.

1.     For evaluating the effects of miRNA-378 expression on XBP1 and Inhibits Growth and Migration of Luminal Breast Cancer Cells, miRNA-378-overexprtession cancer cells and miRNA-378-knowck out cancer cells should all be added in the experiments in vitro of this study. 

2.     Animal study is required. The effects of miRNA-378 on Growth and Migration of Luminal Breast Cancer Cells should be tested in mice model, at least.

3.     While the study emphasizes luminal breast cancer, it would be beneficial to clarify how these findings might differ or be relevant to other breast cancer subtypes, such as triple-negative breast cancer (TNBC), where the role of XBP1 is also noted​​.

4.     (DOI: 10.3736/jcim20120103) is recommended to be cited after “Stressful conditions such as hypoxia, acidosis, oxygen and nutrient deprivation in 41 tumour microenvironment impair the folding capacity of the endoplasmic reticulum 42 (EnR) trigger an evolutionary conserved pathway known as the unfolded protein re-43 sponse (UPR)”.

5.     The paper could provide more detailed mechanistic insights into how miR-378 exerts its regulatory effects on specific target genes, and how these interactions contribute to endocrine resistance in breast cancer.

6.     The manuscript could discuss or speculate on the long-term implications of miR-378 manipulation, particularly in the context of developing resistance to endocrine therapies.

7.     It would be beneficial to discuss potential challenges and considerations in translating these findings into clinical practice, such as the development of miR-378-based therapies or diagnostics. In addition, expanding the discussion to include the potential role of miR-378 and XBP1 in other cancer types or in the context of other cellular stress responses could enrich the manuscript's impact.

Author Response

Reviewer 2

The manuscript titled as "miRNA-378 Is Downregulated by XBP1 and Inhibits Growth and Migration of Luminal Breast Cancer Cells" provides a comprehensive analysis of the role of miRNA-378 in breast cancer, particularly in the context of endoplasmic reticulum (ER) stress and endocrine resistance. Authors concluded that miR-378 upregulates the expression of several genes that regulate type I interferon signalling. Analysis of separate cohorts of breast cancer patients showed that a gene signature derived from miR-378 upregulated genes showed a strong association with improved overall and recurrence free survival in breast cancer. Our results suggest that downregulation of miR-378 by XBP1 contributes to endocrine resistance in ER-positive breast cancer. This study possesses certain novelty, but the evidence should be further enhanced. Thus, major revision is recommended.

  1. For evaluating the effects of miRNA-378 expression on XBP1 and Inhibits Growth and Migration of Luminal Breast Cancer Cells, miRNA-378-overexprtession cancer cells and miRNA-378-knowck out cancer cells should all be added in the experiments in vitro of this study.

Author response: We agree with the reviewer that having data using both gain- and loss-of-function approach will make our arguments more convincing. However, we decided against the loss-of-function approach because our results showed that expression of miR-378 and PPARGC1B was downregulated in tumour samples from breast cancer patients as well as breast cancer cell lines (Fig 1). Reducing the expression of miR-378 in breast cancer cell lines having very low, steady state expression is unlikely to give any meaningful results and we hope that this argument is acceptable.

  1. Animal study is required. The effects of miRNA-378 on Growth and Migration of Luminal Breast Cancer Cells should be tested in mice model, at least.

Author response: Our results show that in miR-378 expressing cells 53 genes were expressed at significantly lower level and 254 were expressed at significantly higher level in miR-378 expressing cells (Fig 8A-B and Table S2). Gene set enrichment analysis of upregulated genes identified significant enrichment of type I interferon signalling pathway genes, while analysis of downregulated genes showed no significant enrichment (SF 7). As such we are not sure about the suitability of xenograft study using MCF7-miR-378 cells in immunodeficient mice. We hope to perform animal studies in immunocompetent syngeneic mice after we replicate the key observations in a murine ER-positive cell line, EO771.   This work is currently ongoing in our group. We believe that - as much as we would like these experiments - in vivo data would be out of the scope of this paper.

  1. While the study emphasizes luminal breast cancer, it would be beneficial to clarify how these findings might differ or be relevant to other breast cancer subtypes, such as triple-negative breast cancer (TNBC), where the role of XBP1 is also noted​​.

Author response: We thank the reviewer for this suggestion. The following text is added to the revised manuscript.

While XBP1 is primarily known for its role in EnR stress and protein folding, emerging evidence suggests its involvement in development and progression of breast cancer. Indeed, several studies have revealed the role of XBP1 in TNBC and ER-positive breast cancers. It has been reported that XBP1s physically interact with Hypoxia Inducible Factor 1α (HIF1α) and c-MYC in context of TNBC [35]. The XBP1s co-operates with HIF1α for optimal expression of HIF1α target genes in TNBC [35]. There is substantial crosstalk between MYC and IRE1-XBP1 axis in TNBC where MYC binds to the promoter region of IRE1 and promotes transcriptional activation of IRE1 leading to enhanced XBP1 splicing [36]. MYC forms a complex with XBP1s and increases its transcriptional activity [36]. IRE1-XBP1 axis regulates secretion of pro-tumourigenic cytokines in TNBC cells [37]. The crosstalk between XBP1s and estrogen signalling creates a positive feedback loop that results in increased expression of XBP1 in ER-positive breast cancer [23]. XBP1s can contribute to endocrine resistance by enhancing the transcriptional activity of point mutants (Y537S, D538G) and fusion mutants (ESR1-YAP1, ESR1-DAB2) of ESR1 [22]. Knockdown of XBP1 in genome-edited MCF7 cells expressing Y537S mutant reduced their growth and re-sensitized them to endocrine therapy [22]. In ER-positive breast cancer XBP1s upregulates the expression of BCL2, p65/RelA, NCOA3, RRM2, CDC6 and TOP2A [24]. Furthermore, NCOA3 RRM2 and CDC6 are mediators of endocrine resistance downstream of XBP1s in ER-positive breast cancer [20, 24]. These studies provide a rationale for combining the XBP1 targeting agents with endocrine therapy to overcome endocrine resistance in breast cancers. Thus, XBP1 can drive tumour progression in TNBC and ER-positive breast cancer via distinct mechanisms due to direct interaction with unique transcription factors and crosstalk with different signaling nodes. Page 18-19 of revised manuscript

  1. (DOI: 10.3736/jcim20120103) is recommended to be cited after “Stressful conditions such as hypoxia, acidosis, oxygen and nutrient deprivation in 41 tumour microenvironment impair the folding capacity of the endoplasmic reticulum 42 (EnR) trigger an evolutionary conserved pathway known as the unfolded protein re-43 sponse (UPR)”.

Author response: We are sorry, but as per the abstract the suggested citation is not relevant to the current manuscript and is in Chinese which we cannot comprehend.

  1. The paper could provide more detailed mechanistic insights into how miR-378 exerts its regulatory effects on specific target genes, and how these interactions contribute to endocrine resistance in breast cancer.

Author response: Surprisingly we found that in miR-378 expressing cells 53 genes were significantly downregulated and 254 were upregulated as compared to control cells. The increased expression is most likely due to an indirect effect of miRNA. Further work is required to identify the targets of miR-378 that modulate the response to endocrine therapy.

  1. The manuscript could discuss or speculate on the long-term implications of miR-378 manipulation, particularly in the context of developing resistance to endocrine therapies.

Author response: We thank the reviewer for this suggestion. The following text is added to the revised manuscript.

The expression of miR-378 has been shown to be downregulated in endocrine resistant models (TamR and LTED cells) of breast cancer and breast cancer tissues [31]. Knockdown of miR-378 has been shown to reverse the tamoxifen-mediated inhibition of cell growth in MCF7 cells [31]. Considering our results, we speculate that downregulation of miR-378 by XBP1s may modulate the endocrine resistance in ER-positive breast cancer. Page 20 of revised manuscript

  1. It would be beneficial to discuss potential challenges and considerations in translating these findings into clinical practice, such as the development of miR-378-based therapies or diagnostics. In addition, expanding the discussion to include the potential role of miR-378 and XBP1 in other cancer types or in the context of other cellular stress responses could enrich the manuscript's im

Author response: We thank the reviewer for the suggestion. We would like to point out that despite the impressive advances in cancer and miRNA research, they are yet to be adopted and used in the routine clinical care of cancer patients, but this is because the translation of candidate biomarkers from bench to bedside is a lengthy and challenging process.  We would prefer to provide succinct focus on the main messages instead of providing extraneous details.

Round 2

Reviewer 1 Report

Comments and Suggestions for Authors

Authors are commended for their revisions. No further comments 

Author Response

Thank you.

Reviewer 2 Report

Comments and Suggestions for Authors

The major concerns have not been addressed. For instance, the hypothesis should be further confirmed by animal models. What is equally important is the animal study can evaluate the safety of the XBP1/miRNA-378 regulation therapy which has not been tested in this study. In addition, miRNA-378-overexprtession and miRNA-378-knowck out cancer cells should also be applied to evaluate the regulating effect of XBP1 on miRNA-378, since this axis was the main topic of this manuscript. Substance evidence is absent. I am afraid that I do not recommend acceptance of this manuscript. 

Author Response

The in vivo data requested is out of the scope of this paper.